# Longevity Regulation by Proline Oxidation in Yeast

**DOI:** 10.3390/microorganisms9081650

**Published:** 2021-08-02

**Authors:** Akira Nishimura, Yuki Yoshikawa, Kazuki Ichikawa, Tetsuma Takemoto, Ryoya Tanahashi, Hiroshi Takagi

**Affiliations:** Division of Biological Science, Graduate School of Science and Technology, Nara Institute of Science and Technology, Nara 630-0192, Japan; nishimura@bs.naist.jp (A.N.); y-yoshikawa@bs.naist.jp (Y.Y.); ichikawa.kazuki.ig4@bs.naist.jp (K.I.); takemoto.tetsuma.to9@bs.naist.jp (T.T.); tanahashi.ryoya.ti3@bs.naist.jp (R.T.)

**Keywords:** chronological lifespan, energy metabolism, homeostasis, longevity, proline, proline oxidase, *Saccharomyces cerevisiae*

## Abstract

Proline is a pivotal and multifunctional amino acid that is used not only as a nitrogen source but also as a stress protectant and energy source. Therefore, proline metabolism is known to be important in maintaining cellular homeostasis. Here, we discovered that proline oxidation, catalyzed by the proline oxidase Put1, a mitochondrial flavin-dependent enzyme converting proline into ∆^1^-pyrroline-5-carboxylate, controls the chronological lifespan of the yeast *Saccharomyces cerevisiae*. Intriguingly, the yeast strain with *PUT1* deletion showed a reduced chronological lifespan compared with the wild-type strain. The addition of proline to the culture medium significantly increased the longevity of wild-type cells but not that of *PUT1*-deleted cells. We next found that induction of the transcriptional factor Put3-dependent *PUT1* and degradation of proline occur during the aging of yeast cells. Additionally, the lifespan of the *PUT3*-deleted strain, which is deficient in *PUT1* induction, was shorter than that of the wild-type strain. More importantly, the oxidation of proline by Put1 helped maintain the mitochondrial membrane potential and ATP production through the aging period. These results indicate that mitochondrial energy metabolism is maintained through oxidative degradation of proline and that this process is important in regulating the longevity of yeast cells.

## 1. Introduction

Aging is a progressive decline in biological functions occurring in all living organisms, ultimately leading to cell death [1]. Organisms are exposed to various environmental stresses (e.g., nutrient starvation, oxidative stress, temperature shift) during their lifespan and aging is associated with homeostatic imbalances in the form of reduced capacity to respond to various stresses [2,3]. Molecules that work to maintain cellular homeostasis are essential for longevity.

Proline is a multifunctional amino acid in organisms. In addition to being a proteogenic amino acid, proline functions as a stress protectant, namely an osmolyte, oxidative stress protectant, protein-folding chaperone, membrane stabilizer and scavenger of reactive oxygen species [4,5,6,7]. In short, proline is known to be important in maintaining cellular homeostasis by protecting cells from various environmental stresses. Thus, the metabolic regulation of proline, including biosynthesis, degradation and transport, has been of great interest. In the budding yeast *Saccharomyces cerevisiae* (*S. cerevisiae*), proline is synthesized from glutamate by three cytoplasmic enzymes, the γ-glutamyl kinase Pro1, the γ-glutamyl phosphate reductase Pro2, and the Δ^1^-pyrroline-5-carboxylate (P5C) reductase Pro3 [8,9]. In most eukaryotes, including yeast, proline is degraded predominantly in mitochondria. The mitochondrial proline oxidase Put1 converts proline into P5C, which is then converted to glutamate by the P5C dehydrogenase Put2 [10,11]. Proline oxidation by Put1 on the mitochondrial inner membrane may result in the generation of electrons that are then donated to the mitochondrial electron transport chain (ETC) through flavine adenine dinucleotide (FAD) to generate ATP [12,13]. *PUT1* and *PUT2* are regulated by the transcriptional activator Put3. Although Put3 is constitutively bound to the *PUT1* and *PUT2* promoters, Put3 is maximally activated for upregulation of *PUT1* and *PUT2* only in the presence of proline and in the absence of preferred nitrogen sources [14]. In addition to the transcriptional regulation of proline metabolisms, excess proline also inhibits the enzymatic activity in a concentration-dependent manner [15]. Unlike bacteria and plants, yeasts do not increase their intracellular proline level under various stress conditions. However, our previous study revealed that the Ile150Thr variant of Pro1 was less sensitive to negative feedback inhibition by proline [15]. Thus, yeast cells accumulate proline by expressing the mutant *PRO1* gene encoding the Ile150Thr variant. Yeast cells with proline accumulation have been shown to exhibit an increase in stress tolerance to freezing, desiccation, oxidation, and ethanol [16].

The aging of yeast cells is measured in two different ways: the replicative lifespan and the chronological lifespan [17]. The replicative lifespan is defined as the number of mitotic divisions before senescence and is used as a model to study the aging of proliferative cells (stem cells, etc.). During the process of mitotic division, yeast cells are exposed to various stresses such as osmotic shock. Therefore, stress protectants in cells are important for replicative lifespan regulation. In fact, we recently reported that intracellular proline levels regulate the replicative lifespan in yeast [18]. In contrast, chronological lifespan is the time that yeast cells survive under non-proliferative conditions (namely, after the stationary phase) and this period models the aging of postmitotic mammalian cells (neuron cells, etc.). After the stationary phase, yeast cells are exposed to nutrient starvation; thus, energy homeostasis in cells is known to be critical for longevity. Basically, the quality and quantity of nutrients in the environment are key factors of longevity regulation. It is common knowledge that calorie restriction extends longevity [19]. The cellular response to calorie restriction occurs partly through nutrient-sensing mechanisms containing the target of rapamycin, cAMP-PKA, and/or the Snf1 pathways [20]. These signals maintain cellular homeostasis by controlling energy metabolism, stress responses and so on, leading to regulation of the lifespan.

In this study, we discovered that Put1 is important for the chronological lifespan of yeast. During the aging period, *PUT1* expression was positively regulated by Put3 and was required for longevity. More importantly, the oxidation of proline by Put1 helped maintain mitochondrial membrane potential and ATP production, leading to an increase in lifespan. Our results suggest a possible mechanism for longevity regulation via proline metabolism in yeast.

## 2. Materials and Methods

### 2.1. Yeast Strains and Medium

The *S. cerevisiae* wild-type (WT) strain with a BY4741 (MATa *his3*Δ*1 leu2*Δ*0 mat15*Δ*0 ura3*Δ*0*) background was used in this study. Strains *put1*Δ, I150T and I150T *put1*Δ were constructed in the previous study [18]. Strain *put3*Δ was obtained from the BY4741-derived deletion library (EUROSCARF, Frankfurt, Germany).

The growth media used were a synthetic complete medium (SC) (2% glucose, and 0.67% yeast nitrogen base with amino acid and ammonium sulfate) and a yeast extract–peptone–dextrose medium (YPD) (2% glucose, 2% peptone and 1% yeast extract). When necessary, 2% agar was added to solidify the medium.

### 2.2. Growth Test

Yeast cells were cultured at 30 °C in SC medium starting from 0.1 of OD_600_. Cell growth was monitored by measuring OD_600_.

### 2.3. Chronological Lifespan Assay

Five single colonies derived from each strain were analyzed, as previously described with slight modification [21]. Yeast cells were cultured in SC medium (at a starting OD_600_ of 0.1) at 30 °C with shaking. Measurement of cell viability began after 72 h of culture (day 0) and continued every 2–6 days by plating a fraction of the culture onto a fresh YPD plate to determine the number of colony-forming units (CFUs). To investigate the effects of the proline supplement, proline (100 µM) was added every 5 days. Cell survival rates were calculated by normalizing the CFUs at each time point to the CFUs at day 0.

### 2.4. Determination of Intracellular Proline Content

Yeast cells were inoculated into SC medium starting from 0.1 of OD_600_. After cultivation at 30 °C for the indicated time in figures under a shaking condition, cells (40 of OD_600_ unit) were collected, resuspended with 1.0 mL of water, and subsequently boiled for 20 min to release amino acids from cells. After centrifugation (5 min at 15,000× *g*), proline content in the supernatant was determined with an amino acid analyzer (JLC-500/V, JEOL, Tokyo, Japan).

### 2.5. Quantitative PCR Analysis

Yeast cells were disrupted by using the Multi-Beads Shocker (Yasui Kikai, Osaka, Japan) with 0.5-mm glass beads, and total RNA was extracted with the NucleoSpin RNA Plus kit (Takara Bio, Shiga, Japan) according to the manufacturer’s instructions. cDNA was synthesized from total RNA with the PrimeScript RT reagent Kit (Takara Bio). The relative abundance of *PUT1* mRNA was quantified by means of quantitative PCR (qPCR) with the a Light Cycler 96 system (Roche, Indianapolis, IN, USA) and SsoAdvanced Universal SYBR Green Supermix (Bio-Rad Laboratories, Hercules, CA, USA). The following primer sets were used in this analysis: *PUT1* qPCR Fw (5′-GGC TGC TAA TCT GAT GGT TGA AA-3′) and Rv (5′-GAG TTA GGA GGT GCC ATC ACA CT-3′), PCR efficiency: 95.3%; and *ACT1* qPCR Fw (5′-CAC CAA CTG GGA CGA TAT GGA-3′) and Rv (5′-GGC AAC TCT CAA TTC GTT GTA GAA-3′), PCR efficiency: 98.5%. The following PCR protocol was used: 95 °C for 4 min followed by 40 cycles of denaturation at 95 °C for 15 s and annealing/extension at 60 °C for 30 s. Each gene’s cycle threshold was normalized to the *ACT1* gene and the relative gene expression compared to WT strain in the log phase was shown, where 1.0 indicated no change in abundance.

### 2.6. Measurement of Mitochondrial Membrane Potential

The mitochondrial membrane potential was measured by using JC-10 (Abcam, Cambridge, UK), a cationic dye that shows potential-dependent accumulation in mitochondria. Yeast cells were cultured in SC medium, harvested by centrifugation, and washed once with 50 mM phosphate buffer. The harvested cells were incubated with JC-10 (10 µM) in 50 mM phosphate buffer at 30 °C for 30 min and then subjected to flow cytometry analysis. Flow cytometry was performed by using a BD Accuri C6 flow cytometer (BD Biosciences, Franklin Lakes, NJ, USA), and the red (FL2)/green (FL1) values were calculated to determine mitochondrial membrane potential.

### 2.7. Measurement of ATP

Yeast cells were cultured in SC medium at 30 °C and were then harvested. The harvested cells of about 2 × 10^7^ cells (OD_600_ = 1.0) were washed three times with ice-cold phosphate buffer (50 mM) and suspended in 200 µL of sterilized water. After cells were frozen with liquid nitrogen, they were heated at 95 °C for 10 min to release ATP into the outside of the cells. ATP levels in 100 µL of supernatant were determined by using the IntraCellular ATP assay kit (Toyo B-net, Tokyo, Japan).

### 2.8. Statistical Analysis

Results are presented as means ± standard deviations (SD) of the indicated experiment numbers in legends. Statistical significance was evaluated by using Student’s *t*-test for two-group comparisons and one-way/two-way ANOVA with Tukey’s test for multiple-group comparisons. These analyses were performed by using GraphPad Prism 7 (GraphPad Software, San Diego, CA, USA). *p* < 0.05 was considered to be statistically significant.

## 3. Results

Our previous study reported that yeast cells with proline accumulation showed tolerance to various stresses (e.g., freezing, desiccation, oxidation, and ethanol) and an extended replicative lifespan [16,18]. However, the effect of proline on the chronological lifespan was still unclear. In our present study, first, to examine the effect of proline metabolism on cell growth, we monitored the OD_600_ of WT and proline oxidase-encoding *PUT1*-deleted (*put1*Δ) strains (Figure 1a). There were no significant differences in the cell growth between the WT and *put1*Δ strains at the beginning of cultivation during the lag and exponential phases. In contrast, the maximal growth in the stationary phase in *put1*Δ was slightly lower than that in WT. To clarify whether the difference in the maximum growth is related to their chronological lifespans, we determined those values for WT and *put1*Δ (Figure 1b). In this study, we evaluated the chronological lifespan as the mean lifespan. Importantly, the chronological lifespan of *put1*Δ was much shorter than that of WT. These results demonstrate that the knockout of *PUT1*, hence the lack of oxidation of proline, contributes to reduce longevity in yeast cells.

We next prepared several yeast strains (WT, *put1*Δ, I150T, and I150T *put1*Δ) with different amounts of accumulated proline to investigate whether the intracellular proline content correlates to the chronological lifespan and/or replicative lifespan [18]. I150T is a mutant expressing *PRO1*^I150T^, which encodes the *γ*-glutamyl kinase variant (Ile150Thr) with desensitization to feedback inhibition by proline. I150T *put1*Δ is a double mutant with both *PRO1*^I150T^ expression and *PUT1* deletion. As expected, the intracellular proline content was higher in the order of I150T *put1*Δ > I150T > *put1*Δ > WT at the stationary phase as reported in a previous study [18] (Figure 2a). The proline level of strain I150T was approximately 2.2-fold higher than that of WT, and that of the I150T *put1*Δ double mutant was almost 1.8-fold higher than that of *put1*Δ. To our surprise, I150T *put1*Δ, which has the highest proline content, showed a shorter lifespan than WT and I150T, but one comparable to that of *put1*Δ (Figure 2b). I150T showed the same lifespan as WT. These results strongly suggest that disruption of *PUT1* causes a decrease in longevity, independent of the intracellular proline content.

To observe the effect of proline supplementation on longevity, we determined chronological lifespans under the condition of adding proline to the culture medium every 5 days. The exogenous proline slightly but significantly extended the lifespan of WT (Figure 3a). On the other hand, proline addition had no effect on the lifespan of *put1*Δ (Figure 3b). Presumably, proline catabolism by the Put1 is involved in the regulation of longevity, but proline itself is directly not.

To verify the importance of Put1 in aging, *PUT1* expression was analyzed by using a qPCR method. Figure 4a shows that *PUT1* was dramatically induced during the aging period. Generally, *PUT1* is positively regulated by the transcriptional factor Put3 [14]. Therefore, we checked *PUT1* induction in *put3*Δ. There was little change of *PUT1* transcription during the lifespan in *put3*Δ, unlike in WT (Figure 4a). Next, changes in intracellular proline were observed in WT, *put1*Δ, and *put3*Δ during the aging period (Figure 4b). In WT, proline content was gradually decreased, whereas proline was hardly changed in *put1*Δ and *put3*Δ. Furthermore, we determined the lifespan of *put3*Δ. Figure 4c shows that the chronological lifespan of *put3*Δ was significantly shorter than that of WT. These results implied that Put3-dependent *PUT1* induction is needed for the proline catabolism during aging, which may regulate the lifespan.

The oxidation of proline by Put1 can generate electrons that are then donated to mitochondrial electron transport to produce ATP. This enzymatic process is likely to play a role in maintaining cellular energy homeostasis. Thus, we finally investigated whether Put1 contributes to mitochondrial energy metabolism. Interestingly, *put1*Δ showed lower mitochondrial membrane potential than WT during the aging period, as assessed by JC-10 fluorescence analysis (Figure 5a). We further measured ATP content during aging (Figure 5b). ATP content in WT was slowly decreased during the aging period, while that in *put1*Δ was drastically reduced compared with WT. This result correlated well with the membrane potentials of WT and *put1*Δ and indicated a decline of energy homeostasis in *put1*Δ.

## 4. Discussion

In this study, we discovered that the proline oxidase Put1 contributes to the chronological lifespan of yeast. Upon yeast cells entering the aging period, *PUT1* expression was positively regulated by Put3 and was associated with longevity. More intriguingly, Put1 helped sustain mitochondrial membrane potential and ATP production, leading to maintenance of the energy metabolism. Our present data suggest a possible mechanism for the longevity regulation via proline metabolism (Figure 6). Yeast cells normally produce ATP from glucose and ethanol via glycolysis in the cytosol and/or ETC in mitochondria, respectively. When all glucose and ethanol are consumed, yeast cells probably uptake proline from the environment and start to utilize proline as an energy source via its oxidation by Put1. This conversion of proline into P5C results in the generation of electrons that are donated to ETC through FAD to generate ATP. The mechanism functions for the homeostasis of energy metabolism, which contributes to longevity.

Proline is a multifunctional amino acid in many organisms. In addition to being a proteogenic amino acid, proline functions as a stress protectant, namely an osmolyte, oxidative stress protectant, protein-folding chaperone, membrane stabilizer and scavenger of reactive oxygen species [4,5,6,7]. Therefore, proline itself may protect yeast cells against various stresses caused by aging and regulate the longevity. However, our present study showed that the amount of proline in the cell does not affect the lifespan. In addition, proline addition had no effect on the lifespan of *put1*Δ cells. These results indicated that the use of proline for ATP synthesis is more important for longevity than as a protective molecule. Aged cells are known to accumulate misfolded proteins or protein aggregates inside the cell due to downregulation of protein quality control systems [22]. Molecular chaperones are central components of such systems and require ATP for protein refolding. Thus, ATP synthesized via proline oxidation catalyzed by Put1 may be used to support the activity of molecular chaperones during aging period, leading to removal of unfolded proteins and prolongation of the longevity. Further studies are needed to reveal that the proline catabolism helps the chaperone activity under aging.

Since yeast cells can utilize various nitrogen sources, the sensing of nitrogen sources is important for optimizing cellular metabolism. Generally, the presence of a preferred nitrogen source, such as ammonium ions or glutamine, represses the expression of catabolic genes compared to nonpreferred nitrogen sources, such as proline and leucine, through nitrogen catabolite repression (NCR) [23,24,25]. All of the proline-metabolizing genes are NCR-regulated genes; that is, their transcription is repressed by the NCR repressors Ure2 and Dal80 when a preferred nitrogen source is present [26]. In addition, the proline transporters Gap1 and Put4 are well-known to be subjected to post-translational regulation, which suppresses their transport activity by the ubiquitin ligase Rsp5 [27,28]. Thus, proline appears to be consumed, similar to other nonpreferred nitrogen sources. In fact, however, yeast cells rarely uptake and consume extracellular proline, even though they do use other nonpreferred nitrogen sources, such as leucine and 4-aminobutanoic acid [29,30,31]. In short, both the uptake and metabolism of proline are strongly inhibited in cells compared with other nonpreferred nitrogen sources. In light of our present study, yeast cells may adopt a strategy of leaving proline outside the cell as a last resort for survival. The presence of energy sources such as glucose strongly inhibits the uptake of proline. In the absence of other energy sources, yeast cells may uptake proline via Gap1 and Put4 and oxidize proline by Put1, which produces ATP to assist in cellular survival. This suggests that proline metabolism is not only regulated by nitrogen sources but also by energy sources such as glucose. Our previous study showed that *PUT1* induction is positively regulated by the general stress transcription factors Msn2/4 [18]. The important fact is that Msn2/4 is activated by the low-glucose sensor Snf1 [32]. Hence, Snf1 may sense energy source depletion in the environment and subsequently Msn2/4 and Put3 may work cooperatively to activate proline metabolism, leading to maintenance of energy homeostasis. Further studies will be needed for understanding the detailed mechanism and physiological significance of proline-mediated energy homeostasis.

In this study, we found that proline taken up from outside the cell, rather than proline synthesized inside the cell, contributes to the longevity of yeast. On the other hand, a previous study indicated that the intracellular proline regulates the replicative lifespan of yeast [18]. Generally, proline is present in the cytosol, but most of excess proline is stored in the vacuoles [33]. Since a mechanism for active transports of proline from the vacuoles to the cytosol has not been identified, proline in the vacuoles may remain there indefinitely. The most severe stress under replicative events is considered to be osmotic stress, and vacuoles are important organelles for the adaptation to osmotic pressure [34]. Thus, proline maintained in vacuoles may be functional during the replicative lifespan. In the case of regulating the chronological lifespan, however, proline must be transported from the cytosol to the mitochondria to serve as a substrate for Put1. Therefore, it is likely to be important for yeast cells to uptake extracellular proline upon entering the aging period.

Proline metabolisms is highly conserved across various organisms. In terms of proline degradation, the localization of this process and the properties of the enzymes involved are very similar between yeasts and mammals. It is known that the mitochondrial proline oxidase is implicated in supporting ATP production, protein synthesis, and redox homeostasis in cancer cells [35]. One of the most lethal capacities of cancer cells is their ability to metastasize to distant organs. Recently, Elia et al. suggested that proline metabolism supports the metastatic cascade via the production of ATP [36]. Importantly, cancer cells in metastatic tissue exhibit higher expression of proline oxidase than normal cells. Accordingly, inhibition of proline oxidase activity impairs metastasis formation in different kinds of metastatic breast cancer [36]. However, the details of how ATP production is involved in metastatic progression are still unknown. Considering our present data, proline oxidase may function as an enzyme to prevent cellular senescence mediated by ATP production. Likewise, inhibition of senescence is well-known to induce cancer formation and metastasis [37]. Thus, an inhibitor of proline oxidase would be a promising candidate as a drug that interferes with the metastatic cascade. Yeast cells can grow on the medium containing proline as a sole nitrogen source, but Put1, an orthologue of proline oxidase, is essential for such growth. Therefore, inhibitors of proline oxidase may be screened by a chemical biology-based study utilizing yeast.

The aging process is a part of the life cycle of all organisms. Yeasts are very important microorganisms for various industrial fields, such as the production of fermented foods and useful compounds. A high fermentative capacity is a key factor for their biotechnological application. Since the yeast performance in biotechnological applications is dependent on levels of cell viability and vitality, the extension of the lifespan is important to maximize these processes. Hence, isolating and/or engineering yeast strains with the appropriate localization and concentration of proline could contribute to improvements in fermentation and other kinds of biochemical productivity.

## Figures and Tables

**Figure 1 microorganisms-09-01650-f001:**
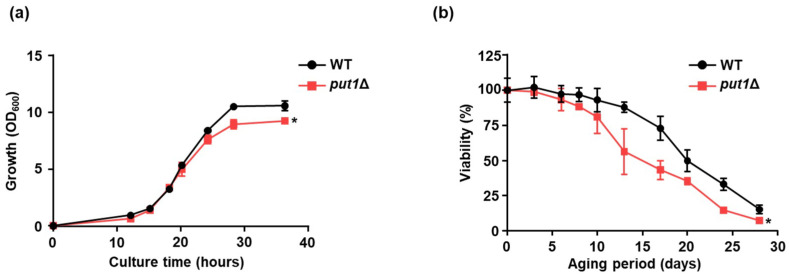
Phenotype analysis of *put1*Δ strain. (**a**) Growth curves of WT and *put1*Δ. Cell growth was determined at the indicated time points by measuring OD_600_. Data are presented as means ± SD (*n* = 3) and statistical significance was determined by two-way ANOVA with Tukey’s test. * *p* < 0.05, vs. WT. (**b**) Chronological survival curves of WT and *put1*Δ. Chronological lifespan was determined by colony formation units. Data are presented as means ± SD (*n* = 5) and statistical significance was determined by two-way ANOVA with Tukey’s test. * *p* < 0.05, vs. WT.

**Figure 2 microorganisms-09-01650-f002:**
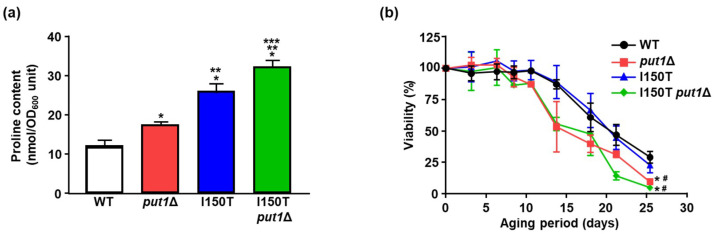
Put1-dependent longevity regulation. (**a**) Proline content in WT, *put1*Δ, I150T, and I150T *put1*Δ. After cultivation for 72 h, proline content was measured with an amino acid analyzer. Data are presented as means ± SD (*n* = 3) and statistical significance was determined by one-way ANOVA with Tukey’s test. * *p* < 0.05, vs. WT; ** *p* < 0.05, vs. *put1*Δ; *** *p* < 0.05, vs. I150T. (**b**) Chronological survival curves of WT, *put1*Δ, I150T and I150T *put1*Δ. Chronological lifespan was determined by counting the number of colony-forming units. Data are presented as means ± SD (*n* = 5) and statistical significance was determined by two-way ANOVA with Tukey’s test. * *p* < 0.05, vs. WT; ^#^
*p* < 0.05, vs. I150T.

**Figure 3 microorganisms-09-01650-f003:**
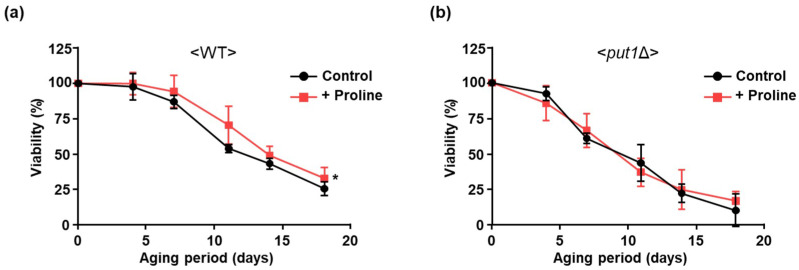
Effect of proline supplement on longevity. Chronological survival curves of WT (**a**) and *put1*Δ (**b**) cells supplemented with proline (+ Proline) were determined by counting the number of colony-forming units. Proline (100 µM) was added to medium every 5 days. Data are presented as means ± SD (*n* = 5) and statistical significance was determined by two-way ANOVA with Tukey’s test. * *p* < 0.05, vs. non-treated control (Control).

**Figure 4 microorganisms-09-01650-f004:**
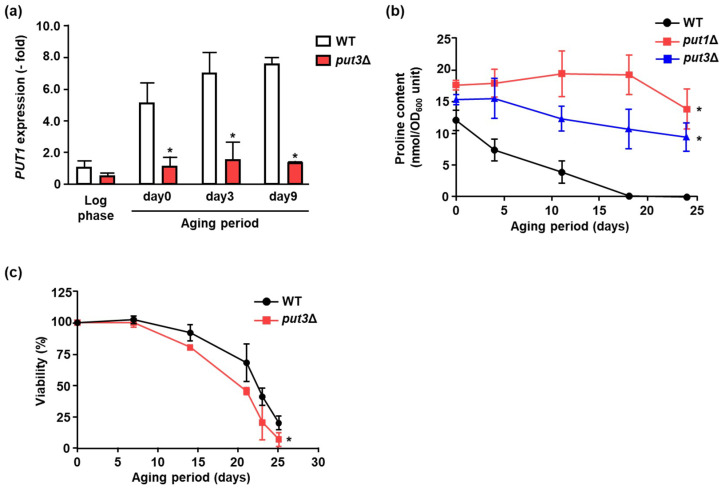
Longevity regulation mediated by *PUT1* induction. (**a**) *PUT1* expression in WT and *put3*Δ upon aging period. *PUT1* mRNA was determined by qPCR. Cells in log phase were collected after the cultivation for 12 h. Data are presented as means ± SD (*n* = 3) and statistical significance was determined by Student’s *t*-test. * *p* < 0.05, vs. WT. (**b**) Proline content in WT, *put1*Δ and *put3*Δ upon aging period. After cultivation for the indicated time, proline content was measured by using an amino acid analyzer. Data are presented as means ± SD (*n* = 3) and statistical significance was determined by two-way ANOVA with Tukey’s test. * *p* < 0.05, vs. WT. (**c**) Chronological survival curves of WT, *put3*Δ. Chronological aging was determined by counting the number of colony-forming units. Data are presented as means ± SD (*n* = 5) and statistical significance was determined by two-way ANOVA with Tukey’s test. * *p* < 0.05, vs. WT.

**Figure 5 microorganisms-09-01650-f005:**
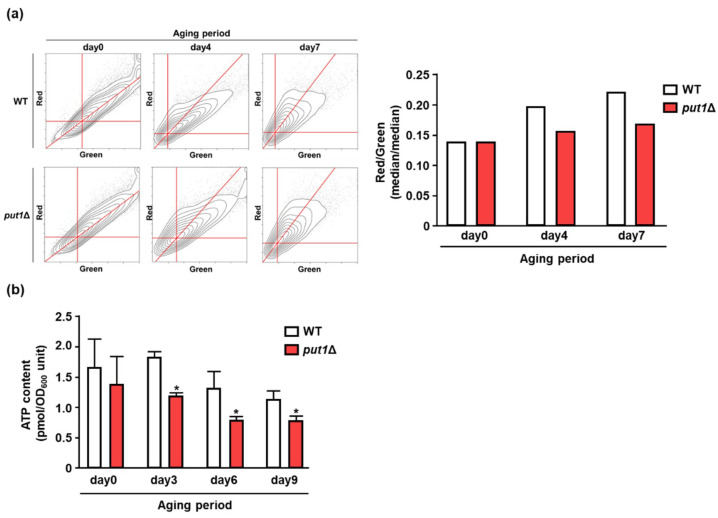
Put1-dependent mitochondrial energy production. (**a**) Mitochondrial membrane potential in WT and *put1*Δ upon aging period. Left panel indicates representative contour-plot of JC-10 flow cytometry (event counts = 10,000). The intersection of the lines indicates the maximum population. Right panel shows median of fluorescence ratios, JC-10 aggregates (red) vs. JC-10 monomers (green). The viability of the cells used in this assay was 80–100%. Three independent experiments were performed, but representative results are presented here due to the large variation in absolute values. (**b**) ATP content in WT and *put1*Δ upon aging period. ATP was detected by using the luciferase assay. Data are presented as means ± SD (*n* = 10) and statistical significance was determined by Student’s *t*-test. * *p* < 0.05, vs. WT.

**Figure 6 microorganisms-09-01650-f006:**
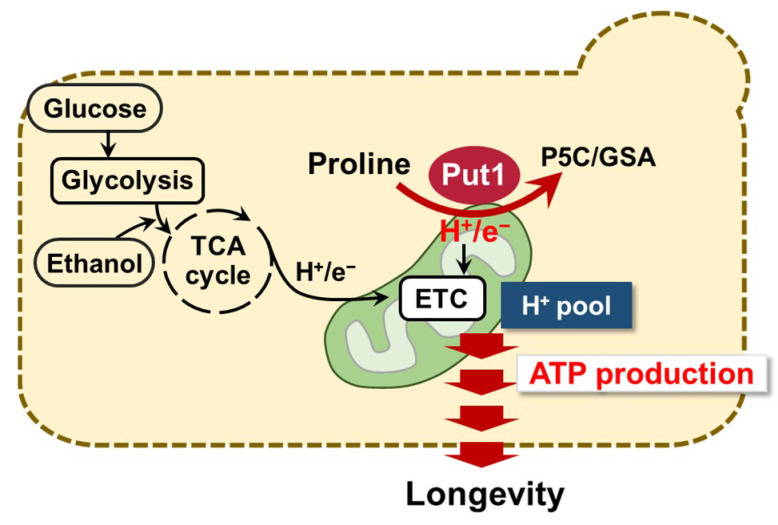
Schematic model of longevity regulation by proline oxidation. In the absence of principal energy sources, such as glucose and ethanol, yeast cells start to utilize proline as an energy source via the Put1-catylatic reaction. This reaction results in the generation of electrons which are donated to ETC, which generates ATP. Such a mechanism may contribute to longevity.

## Data Availability

The analyzed data presented in this study are included within this article. Further data is available on reasonable request from the corresponding author.

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
