# Peer review of "Longevity Regulation by Proline Oxidation in Yeast"

_microorganisms, 2021, doi:10.3390/microorganisms9081650_

Round 1

Reviewer 1 Report

In the Manuscript „Longevity regulation by proline oxidation in yeast“ Akira Nishimura et al describe how the proline oxidase Put1 influences the longevity of the yeast S. cerevisiae. The authors discovered that the removal of Put1 causes a chronological lifespan decrease, and disrupts the mitochondrial membrane potential and ATP synthesis. 
Despite the experiments are well performed, some of the drawn conclusions appear contradictory and too strong for the presented results. 

Comments:

- The growth curve in Figure 1 would be better represented in semi-logarithmic scale.

- The yeast was grown on fermentable carbon sources. Did the authors test how growth and ATP synthesis of the Put1 knockout strain and of the other investigated strains is affected on non-fermentable carbon sources? This experiment would add information on how the energy metabolism is influenced by proline.

- 170-171: The statement that the proline metabolism regulates longevity is too strong and not demonstrated by the experiments presented in Fig 1. The results shown here demonstrate that the knockout of PUT1, hence the lack of oxidation of proline, is contributing to reduce longevity. In fact, the author state that the intracellular abundance of proline is not directly correlating with longevity (Fig 2). 

- 192-193: If „the disruption of PUT1 causes a decrease in longevity independent of the intracellular proline content“ could the authors improve their discussion on how the PUT1 knockout might influence longevity else wise? 

- 209-210: this statement is very vague. If the addition of proline is not enhancing PUT1 knockout growth then this strain might have an issue with proline utilisation. Also, this statement contradicts what was said in lines 170-171.

- 227-229: This statement is very strong and contradicts what was said in lines 192-194 and 209-210. If proline consumption is indeed regulating the lifespan then the longevity should correlate with the proline content.

- 270-271: Is the putative link between proline and longevity due to its usage for ATP synthesis or by its capability to act as a protective molecule against cellular stress? The authors should improve discussion on this aspect

- 295-296: How do yeast cells store proline outside of the cell?

- 308-309: Is the extracellular proline not taken up by the cells and becomes then intracellular?

Author Response

Reviewer: 1

In the Manuscript “Longevity regulation by proline oxidation in yeast” Akira Nishimura et al describe how the proline oxidase Put1 influences the longevity of the yeast S. cerevisiae. The authors discovered that the removal of Put1 causes a chronological lifespan decrease, and disrupts the mitochondrial membrane potential and ATP synthesis. Despite the experiments are well performed, some of the drawn conclusions appear contradictory and too strong for the presented results.

Comments:

  1. The growth curve in Figure 1 would be better represented in semi-logarithmic scale.

According to your suggestion, we recreated Figure 1a with semi-logarithmic scale. However, the recreated version (please see the right side of figure below) made it difficult to observe the difference between WT and put1Δ strains. Therefore, we have decided to use the original version of Figure 1a in this paper.

 (a) Liner graph (original version), (b) Semi-logarithmic graph (recreated version).

  1. The yeast was grown on fermentable carbon sources. Did the authors test how growth and ATP synthesis of the Put1 knockout strain and of the other investigated strains is affected on non-fermentable carbon sources? This experiment would add information on how the energy metabolism is influenced by proline.

Your suggestion is an interesting perspective, but we do not think that experiments with non-fermentable carbon sources are necessary in this study. Since non-fermentable carbon enhances the mitochondrial activity, the cells cultured on non-fermentable carbon sources may show the high activity of proline catabolism even before the aging period. However, our present study suggest that the proline catabolism just during the aging of yeast cells is important for the energy metabolism and longevity. Thus, we think that experiments with non-fermentable carbon sources does not clarify the effect of proline.

  1. 170-171: The statement that the proline metabolism regulates longevity is too strong and not demonstrated by the experiments presented in Fig 1. The results shown here demonstrate that the knockout of PUT1, hence the lack of oxidation of proline, is contributing to reduce longevity. In fact, the author state that the intracellular abundance of proline is not directly correlating with longevity (Fig 2).

We have agreed your keen suggestion that the sentence is too strong. Therefore, we have changed the sentence “These results indicated that proline metabolism regulates longevity in yeast cells” to “These results demonstrate that the knockout of PUT1, hence the lack of oxidation of proline, contributes to reduce longevity in yeast cells” (P. 4, L. 172-173).

  1. 192-193: If „the disruption of PUT1 causes a decrease in longevity independent of the intracellular proline content “could the authors improve their discussion on how the PUT1 knockout might influence longevity elsewise?

Aged cells are known to accumulate misfolded proteins or protein aggregates inside the cell due to downregulation of protein quality control systems (Krisko and Radman, Open Biol., 2019). Molecular chaperones are central components of such systems and require ATP for protein refolding. Thus, ATP synthesized via proline oxidation catalyzed by Put1 may be used to support the activity of molecular chaperones during aging period, leading to removal of unfolded proteins and prolongation of the longevity. Further studies are needed to reveal that the proline catabolism helps the chaperone activity under aging. We have now included these statements in the revised text (P. 7-8, L. 286-292; P. 10. L. 419).

  1. 209-210: this statement is very vague. If the addition of proline is not enhancing PUT1 knockout growth then this strain might have an issue with proline utilisation. Also, this statement contradicts what was said in lines 170-171.

As you pointed out, the statement might be incorrect. We have changed the sentence “Presumably, the enzymatic activity of Put1 but not directly proline itself is involved in the regulation of longevity.” to “Presumably, proline catabolism by the Put1 is involved in the regulation of longevity, but proline itself is directly not.” (P. 5, L. 211-212).

  1. 227-229: This statement is very strong and contradicts what was said in lines 192-194 and 209-210. If proline consumption is indeed regulating the lifespan then the longevity should correlate with the proline content.

We accepted your opinion and have thus changed the sentence “These results demonstrated that Put3-dependent PUT1 induction is needed for the proline consumption during aging, which may regulate the lifespan.” to “These results implied that Put3-dependent PUT1 induction is needed for the proline catabolism during aging, which may regulate the lifespan.” (P. 5-6, L. 229-231).

  1. 270-271: Is the putative link between proline and longevity due to its usage for ATP synthesis or by its capability to act as a protective molecule against cellular stress? The authors should improve discussion on this aspect.

Proline is a multifunctional amino acid in many organisms. In addition to being a proteogenic amino acid, proline functions as a stress protectant, namely an osmolyte, oxidative stress protectant, protein-folding chaperone, membrane stabilizer and scavenger of reactive oxygen species. Therefore, proline itself may protect yeast cells against various stresses caused by aging and regulate the longevity. However, our present study showed that the amount of proline in the cell does not affect the lifespan. In addition, proline addition had no effect on the lifespan of put1Δ cells. These results indicated that the use of proline for ATP synthesis is more important for longevity than as a protective molecule. These statements are now included in the revised text (P. 7, L. 278-286).

  1. 295-296: How do yeast cells store proline outside of the cell?

The statement might be biased and inconsistent. In order to prevent the reader from misunderstanding, we have simply modified the sentence, from “In light of our present study, yeast cells may adopt a strategy of storing proline outside the cell as a last resort for survival.” to “In light of our present study, yeast cells may adopt a strategy of leaving proline outside the cell as a last resort for survival.” (P. 8, L. 313-314).

  1. 308-309: Is the extracellular proline not taken up by the cells and becomes then intracellular?

Thank you for pointing out the mistake in the sentence. To avoid misunderstanding for readers, we have modified the sentence, from “In the present study, we found that the presence of extracellular but not intracellular proline contributes to the longevity of yeast.” to “In this study, we found that proline taken up from outside the cell, rather than proline synthesized inside the cell, contributes to the longevity of yeast.” (P8. L. 326-327)

Reviewer: 1

In the Manuscript “Longevity regulation by proline oxidation in yeast” Akira Nishimura et al describe how the proline oxidase Put1 influences the longevity of the yeast S. cerevisiae. The authors discovered that the removal of Put1 causes a chronological lifespan decrease, and disrupts the mitochondrial membrane potential and ATP synthesis. Despite the experiments are well performed, some of the drawn conclusions appear contradictory and too strong for the presented results.

Comments:

  1. The growth curve in Figure 1 would be better represented in semi-logarithmic scale.

According to your suggestion, we recreated Figure 1a with semi-logarithmic scale. However, the recreated version (please see the right side of figure below) made it difficult to observe the difference between WT and put1Δ strains. Therefore, we have decided to use the original version of Figure 1a in this paper.

 (a) Liner graph (original version), (b) Semi-logarithmic graph (recreated version).

  1. The yeast was grown on fermentable carbon sources. Did the authors test how growth and ATP synthesis of the Put1 knockout strain and of the other investigated strains is affected on non-fermentable carbon sources? This experiment would add information on how the energy metabolism is influenced by proline.

Your suggestion is an interesting perspective, but we do not think that experiments with non-fermentable carbon sources are necessary in this study. Since non-fermentable carbon enhances the mitochondrial activity, the cells cultured on non-fermentable carbon sources may show the high activity of proline catabolism even before the aging period. However, our present study suggest that the proline catabolism just during the aging of yeast cells is important for the energy metabolism and longevity. Thus, we think that experiments with non-fermentable carbon sources does not clarify the effect of proline.

  1. 170-171: The statement that the proline metabolism regulates longevity is too strong and not demonstrated by the experiments presented in Fig 1. The results shown here demonstrate that the knockout of PUT1, hence the lack of oxidation of proline, is contributing to reduce longevity. In fact, the author state that the intracellular abundance of proline is not directly correlating with longevity (Fig 2).

We have agreed your keen suggestion that the sentence is too strong. Therefore, we have changed the sentence “These results indicated that proline metabolism regulates longevity in yeast cells” to “These results demonstrate that the knockout of PUT1, hence the lack of oxidation of proline, contributes to reduce longevity in yeast cells” (P. 4, L. 172-173).

  1. 192-193: If „the disruption of PUT1 causes a decrease in longevity independent of the intracellular proline content “could the authors improve their discussion on how the PUT1 knockout might influence longevity elsewise?

Aged cells are known to accumulate misfolded proteins or protein aggregates inside the cell due to downregulation of protein quality control systems (Krisko and Radman, Open Biol., 2019). Molecular chaperones are central components of such systems and require ATP for protein refolding. Thus, ATP synthesized via proline oxidation catalyzed by Put1 may be used to support the activity of molecular chaperones during aging period, leading to removal of unfolded proteins and prolongation of the longevity. Further studies are needed to reveal that the proline catabolism helps the chaperone activity under aging. We have now included these statements in the revised text (P. 7-8, L. 286-292; P. 10. L. 419).

  1. 209-210: this statement is very vague. If the addition of proline is not enhancing PUT1 knockout growth then this strain might have an issue with proline utilisation. Also, this statement contradicts what was said in lines 170-171.

As you pointed out, the statement might be incorrect. We have changed the sentence “Presumably, the enzymatic activity of Put1 but not directly proline itself is involved in the regulation of longevity.” to “Presumably, proline catabolism by the Put1 is involved in the regulation of longevity, but proline itself is directly not.” (P. 5, L. 211-212).

  1. 227-229: This statement is very strong and contradicts what was said in lines 192-194 and 209-210. If proline consumption is indeed regulating the lifespan then the longevity should correlate with the proline content.

We accepted your opinion and have thus changed the sentence “These results demonstrated that Put3-dependent PUT1 induction is needed for the proline consumption during aging, which may regulate the lifespan.” to “These results implied that Put3-dependent PUT1 induction is needed for the proline catabolism during aging, which may regulate the lifespan.” (P. 5-6, L. 229-231).

  1. 270-271: Is the putative link between proline and longevity due to its usage for ATP synthesis or by its capability to act as a protective molecule against cellular stress? The authors should improve discussion on this aspect.

Proline is a multifunctional amino acid in many organisms. In addition to being a proteogenic amino acid, proline functions as a stress protectant, namely an osmolyte, oxidative stress protectant, protein-folding chaperone, membrane stabilizer and scavenger of reactive oxygen species. Therefore, proline itself may protect yeast cells against various stresses caused by aging and regulate the longevity. However, our present study showed that the amount of proline in the cell does not affect the lifespan. In addition, proline addition had no effect on the lifespan of put1Δ cells. These results indicated that the use of proline for ATP synthesis is more important for longevity than as a protective molecule. These statements are now included in the revised text (P. 7, L. 278-286).

  1. 295-296: How do yeast cells store proline outside of the cell?

The statement might be biased and inconsistent. In order to prevent the reader from misunderstanding, we have simply modified the sentence, from “In light of our present study, yeast cells may adopt a strategy of storing proline outside the cell as a last resort for survival.” to “In light of our present study, yeast cells may adopt a strategy of leaving proline outside the cell as a last resort for survival.” (P. 8, L. 313-314).

  1. 308-309: Is the extracellular proline not taken up by the cells and becomes then intracellular?

Thank you for pointing out the mistake in the sentence. To avoid misunderstanding for readers, we have modified the sentence, from “In the present study, we found that the presence of extracellular but not intracellular proline contributes to the longevity of yeast.” to “In this study, we found that proline taken up from outside the cell, rather than proline synthesized inside the cell, contributes to the longevity of yeast.” (P8. L. 326-327).

Reviewer 2 Report

In their study, Nishimura et al show that proline oxidation is involved in prolongation of yeast lifespan. 

I have following comments that should be addressed by the authors in a revision:

Fig 1b: The graph does not really show that the chronological lifespan is much shorter. If the authors want to describe the mean lifespan, they should clarify this in the text. At day 27, WT and mutant seemed to reach a comparable viability again.

Fig 3a: Again, the maximum lifespan per se does not seem to be changed

  1. 247: As these dyes for mitochondrial transmembrane potential tend to stain other membranes as well under distinct circumstances, the authors should show a respective microscopy with it (at least representative micrographs).

Fig. 5a: As this dye should emit red fluorescence when inside mitochondria and green fluorescence when cells are apoptotic, I think the authors should re-evaluate their flow cytometry.

Either a compensation method should be applied to avoid bleeding into the other fluorescent channel or at least positive and negative controls should be added here.

Fig 5b: Can the authors mimic the lifespan prolongation by external addition of ATP?

  1. 297: Can the authors prove this statement by simply generating a (conditional) Gap1 or Put4 deletion?

  1. 308: Can the authors simultaneously assess external and internal proline concentrations to validate this statement?

Author Response

Reviewer: 2

In their study, Nishimura et al show that proline oxidation is involved in prolongation of yeast lifespan. I have following comments that should be addressed by the authors in a revision:

Comments:

  1. Fig 1b: The graph does not really show that the chronological lifespan is much shorter. If the authors want to describe the mean lifespan, they should clarify this in the text. At day 27, WT and mutant seemed to reach a comparable viability again.

As you kindly pointed out, we evaluated the chronological lifespan as the mean lifespan. From the standpoint of the mean lifespan, there is a clear difference between the chronological lifespan of WT and put1Δ cells. We have now added the sentence about the definition of the chronological lifespan into the revised text (P. 4, L. 170-171).

  1. Fig 3a: Again, the maximum lifespan per se does not seem to be changed

As the above response, we have now added the sentence about the definition of the chronological lifespan into the revised text (P. 4, L. 170-171).

  1. 247: As these dyes for mitochondrial transmembrane potential tend to stain other membranes as well under distinct circumstances, the authors should show a respective microscopy with it (at least representative micrographs).

We understand your concern that other membranes besides mitochondria possibly was stained by JC10. However, we have not yet performed the analysis with a microscopy. A cationic dye JC10 exhibits fluorescence emission at two typical wavelengths (both excited at 488 nm): (i) red fluorescent J-aggregates (emission maximum at 590 nm) reflecting higher mitochondrial potential; and (ii) green fluorescent J-monomers (emission maximum at 530 nm) indicating lower membrane potential. The ratio of the red/green fluorescence depends only on the mitochondrial membrane potential. In other words, the results that we could detect the change of both red and green fluorescence intensities strongly indicate that JC10 is strictly and mostly localized in mitochondria. Therefore, we believe that the analysis with a microscopy is unnecessary in this study. We would really appreciate it if you would kindly understand our response.

  1. Fig. 5a: As this dye should emit red fluorescence when inside mitochondria and green fluorescence when cells are apoptotic, I think the authors should re-evaluate their flow cytometry.

As you pointed out, apoptotic cells are well-known to show the high level of green autofluorescence. For this reason, the results of analysis of samples with many dead cells are often complicated. However, most of the cells used in the JC10 analysis were viable cells. In fact, the viability of the cells in the samples was 80%-100%. Therefore, we have not re-tested our data. We have now included the sentence regarding the cell viability in the samples in the legend of Figure 5a (P7. L. 260).

  1. Either a compensation method should be applied to avoid bleeding into the other fluorescent channel or at least positive and negative controls should be added here.

We comprehend your concerns about spectral overlap, but our experiments do not raise the concern. BD Accuri C6 flow cytometers used in this study are the most widely cited flow cytometers and renowned for its sensitivity and reliability in the world. Moreover, we performed all JC10 analysis after a process of fluorescence compensation with standard beads. Fluorescence compensation entirely ensures that the fluorescence detected in a particular detector derives from the fluorochrome that is being measured. Thus, spectral overlap is unlikely to occur in our experiments.

  1. Fig 5b: Can the authors mimic the lifespan prolongation by external addition of ATP?

We are interested in experiments which you proposed. However, yeast cells can rarely uptake external ATP across the membrane. Although permeabilized cells treated with detergents can uptake ATP, such cells are unable to be used for lifespan analysis. Thus, we have not yet observed the effect of the ATP addition.

  1. 297: Can the authors prove this statement by simply generating a (conditional) Gap1 or Put4 deletion?

As you pointed out, we would also be interested in verifications of our hypothesis. However, we have not yet obtained enough data to show you and readers. We are currently constructing various strains with PUT4 disruption, conditional and constitutive PUT4 expression, or expression of PUT4 mutant with the deletion of ubiquitination sites. We believe that our hypothesis can be verified by measuring the amount of proline inside and outside the mutant cells and the lifespan of these mutants. We will report these details in the near future. We would really appreciate it if you would kindly understand our response.

  1. 308: Can the authors simultaneously assess external and internal proline concentrations to validate this statement?

We can simultaneously measure both the external and internal proline contents. Since we recognize verifications of the hypothesis as a topic for further research as mentioned in the question above, we have not included the data in this manuscript. We promise that we will reveal the details in the next paper. We would appreciate your kind understanding.

Round 2

Reviewer 1 Report

No further comments

Reviewer 2 Report

While I do not fully agree with all points the authors wrote about in their rebuttal letter (especially regarding representative microscopy and flow cytometry measurements), I accept the manuscript in its current form.

However, I would still emphasise the necessity of providing representative micrographs (for future studies).